# Context-faithful Prompting for Large Language Models

**Wenxuan Zhou[1], Sheng Zhang[2], Hoifung Poon[2], Muhao Chen[1]**
[1]University of Southern California [2]Microsoft Research
[1]{zhouwenx,muhaoche}@usc.edu [2]{zhang.sheng,hoifung}@microsoft.com

## Abstract

Large language models (LLMs) encode parametric knowledge about world facts and have shown remarkable performance in knowledge-driven NLP tasks. However, their reliance on parametric knowledge may cause them to overlook contextual cues, leading to incorrect predictions in context-sensitive NLP tasks (e.g., knowledge acquisition tasks). In this paper, we seek to assess and enhance LLMs' contextual faithfulness in two aspects: knowledge conflict and prediction with abstention. We demonstrate that LLMs' faithfulness can be significantly improved using carefully designed prompting strategies. In particular, we identify opinion-based prompts and counterfactual demonstrations as the most effective methods. Opinion-based prompts reframe the context as a narrator's statement and inquire about the narrator's opinions, while counterfactual demonstrations use instances containing false facts to improve faithfulness in knowledge conflict situations. Neither technique requires additional training. We conduct experiments on three datasets of two standard NLP tasks, machine reading comprehension and relation extraction, and the results demonstrate significant improvement in faithfulness to contexts.[1]

## 1 Introduction

Large language models (LLMs; Brown et al. 2020; Wei et al. 2022; Chowdhery et al. 2022; Chung et al. 2022) have made remarkable advances in solving various NLP problems, particularly in (context-free) knowledge-driven tasks such as question answering (Joshi et al., 2017; Kwiatkowski et al., 2019) and commonsense reasoning (Clark et al., 2018; Mihaylov et al., 2018). Without external context, LLMs can answer factual questions and achieve comparable results to supervised approaches (Brown et al., 2020; Wei et al., 2022), in-

---

[1]Code and data are released at https://github.com/wzhouad/context-faithful-llm.

**Knowledge Conflict**

**Prediction with Abstention**

Figure 1: Examples of knowledge conflict and prediction with abstention. LLMs may ignore the provided context and make unfaithful predictions based on their parametric knowledge before Q4 2021.

dicating that LLMs encode *parametric knowledge* about open-world facts.

Although parametric knowledge can be beneficial for knowledge-driven tasks, overly relying on it can cause problems in context-specific NLP tasks. First, LLMs may encode misconceptions (Lin et al., 2022) or obsolete facts (Lazaridou et al., 2021; Liska et al., 2022; Kasai et al., 2022), in which case we expect LLMs to update their predictions when provided with relevant context. Second, when using LLMs for knowledge acquisition tasks such as machine reading comprehension (MRC; Clark et al. 2019; Rajpurkar et al. 2016) and information extraction (IE; Sang and De Meulder 2003; Zhang et al. 2017; Zhou and Chen 2022; Lu et al. 2022), LLMs should always extract the *knowledge in context* instead of relying solely on their parametric knowledge. In such context-specific application scenarios, we expect LLMs to make decisions faithful to the context and avoid simply parroting answers from pretraining. However, studies have discovered that LLMs can overlook or ignore context (Kasai et al., 2022; Li et al., 2022; Si et al.,

2023), posing a significant challenge for their application in these scenarios.

In this paper, we aim to investigate techniques for improving the faithfulness of LLMs in context-specific NLP tasks. Conceptually, faithfulness is not simply about how much accuracy the model can offer. Instead, it should concern the validity and reliability of its extraction process. Specifically, when there is decision-related information (e.g., a concept or a relation) to extract, a faithful LLM should *genuinely* induce what is described in the context but not give *trivial guesses* based on parametric knowledge or statistical biases. Besides, when no known decision-related information is described in the context, the model should *selectively* abstain from predicting. Accordingly, to provide a realistic assessment of LLMs in terms of faithfulness, we narrow our focus to two sub-problems, namely entity-based knowledge conflict (Longpre et al., 2021; Wang et al., 2022) and prediction with abstention (Rajpurkar et al., 2018), examples of which are shown in Fig. 1. In cases of knowledge conflict, where the given context contains facts different from the pretraining data, LLMs need to return the facts locally described in the context instead of the globally memorized ones. For example, in Fig. 1, text-davinci-003 identifies *Jack Dorsey* instead of *Elon Musk* as the CEO of Twitter, based on its pretrained data before Q4 2021. In cases of prediction with abstention, where the context does not provide information to answer the questions, LLMs should abstain from making predictions and notify the users, rather than answering the questions that become a trivial guess. For example, in Fig. 1, when asked about the founder of Microsoft based on an irrelevant context, LLMs should admit that, from here, they cannot infer the answer.

We present various prompting strategies to improve the faithfulness of LLMs, including designing effective prompts and choosing appropriate in-context demonstrations. We find that constraining the scope of questions to the context by adding phrases (e.g., based on the given context) or natural language instructions improve faithfulness in both facets. Particularly, we find that reformulating the context and questions to opinion-based question-answering problems (Gupta et al., 2019; Bjerva et al., 2020), where the context is expressed in terms of a narrator's statement, and the question asks about this narrator's opinion, delivers the most gains. Additionally, we find that adding counter-

factual demonstrations to prompts improves faithfulness in the aspect of knowledge conflict, while using the original (factual) demonstrations leads to limited or negative effects. Finally, combining both techniques delivers the largest gain than using each one independently.

We evaluate our methods based on three datasets, including Re-TACRED (Stoica et al., 2021) for relation extraction, and natural questions (Kwiatkowski et al., 2019) and RealTime QA (Kasai et al., 2022) for MRC. We find that the proposed strategies can largely improve faithfulness, e.g., reducing the memorization ratio[2] of text-davinci-003 from 35.2% to 3.0% on natural questions. Additionally, we evaluate our methods across LLMs of different scales, finding that larger LLMs are more likely to update memorized answers than smaller ones, both with and without the application of our methods.

## 2 Related Work

We discuss two topics of related work that are closely relevant to this work.

**Knowledge conflicts.** LLMs (Brown et al., 2020; Wei et al., 2022; Chowdhery et al., 2022) have shown promising results in closed-book QA tasks, indicating their ability to memorize facts about the world. However, as the world is constantly evolving, memorized facts may become outdated (Lazaridou et al., 2021; Liska et al., 2022; Kasai et al., 2022), emphasizing the need to update LLMs' predictions with new facts. To address this challenge, some studies (Zhu et al., 2020; De Cao et al., 2021; Mitchell et al., 2022; Meng et al., 2022, 2023) have explored ways to identify and edit the facts stored in model parameters. However, it remains unclear whether memory editing methods allow sufficient capacity to encompass all new factual knowledge. Another promising direction is to augment LLM prompting with external context containing relevant knowledge (Lazaridou et al., 2022; Izacard et al., 2022; Khattab et al., 2022). Coupled with retrieval systems (Karpukhin et al., 2020; Santhanam et al., 2022; Gao and Callan, 2022), such methods have the potential to update LLMs with large amounts of new facts. However, such methods face the challenge that LLMs may persist with the memorized facts and ignore the provided context (Longpre et al., 2021). To tackle this challenge, recent works (Neeman et al., 2022; Li et al., 2022) fine-

---

[2]The percentage of times that LLMs return memorized answers versus answers in the context (Longpre et al., 2021).

tune LLMs on counterfactual contexts, where the original facts are replaced with counterfactual ones. They find that such finetuning processes can effectively improve the LLMs' utilization of contexts. In this study, we propose a novel approach using prompting to improve context faithfulness in LLMs without additional finetuning, which offers a more general and cost-effective method for LLMs.

**Prediction with abstention.** Selective prediction with abstention (Chow, 1970; Fumera and Roli, 2002; Cortes et al., 2016) is an important problem in trustworthy AI. When models are uncertain about their predictions, it is critical that they should admit the uncertainty instead of returning incorrect predictions. Selective prediction may be adopted in different scenarios, such as when instances are close to the decision boundary (Gal and Ghahramani, 2016; Lakshminarayanan et al., 2017; Xin et al., 2021), or when instances are from different domains to training (Hendrycks and Gimpel, 2017; Hendrycks et al., 2020; Zhou et al., 2021). In the scope of context-specific NLP, abstention is preferred when the context is irrelevant to the question. For example, SQuAD 2.0 (Rajpurkar et al., 2018) introduces unanswerable questions to extractive MRC, while Yatskar (2019) finds it focused on questions of extreme confusion and thus is less relevant to the focus of our study. CoQA (Reddy et al., 2019) and QuAC (Choi et al., 2018) introduce unanswerable questions to conversational question answering. RealTime QA (Kasai et al., 2022) finds that GPT-3 still generates outdated answers when provided with irrelevant documents. To address the problem, Neeman et al. (2022) propose the answerability augmentation where LLMs should predict *Unanswerable* when presented with an empty or randomly sampled document. Several other work employ variants of confidence calibration techniques to encourage the NLP model to avoid giving a high confidence on any decisions when encountering a case to abstain (Wang et al., 2023, 2022), which however request white-box accessibility of the incorporated models. We tackle this problem with a part of our prompting method, which we find to significantly enhance the LLMs' ability to make selective predictions without need re-calibration or white-box accessibility of the model.

## 3 Method

We focus on context-specific NLP tasks. The input of these tasks is formulated as $(c, q)$ for free-form

generation tasks, where $c$ is the context and $q$ is the question, or $(c, q, o)$ for tasks with close decision spaces (e.g., multi-choice tasks), where $o$ is the set of decisions/choices. The desired output can be either a free-form text or a choice. We solve these tasks by prompting LLMs and study ways of designing prompting templates and demonstrations that are dedicated to improving the faithfulness of LLMs. Specifically, we find two proposed methods, opinion-based prompts and counterfactual demonstrations, to be the most effective ones. Our methods only change the prompts without finetuning the LLMs (Longpre et al., 2021; Li et al., 2022; Neeman et al., 2022), targeting a more general and affordable solution.

### 3.1 Opinion-based Prompting

Given an input $(c, q, o)$, we begin with the following *base* prompting template:[3]

> **Base prompt**
>
> {$c$} Q: {$q$}? Options: {$o$} A:

Here, {.} serves as a placeholder to be filled with specific content during prompting. We investigate two types of prompting templates for context-specific NLP, namely *opinion-based* prompts and *instructed* prompts. Opinion-based prompts transform original questions into opinion-seeking questions, which naturally demand more attention to the context. Instructed prompts, on the other hand, explicitly instruct LLMs to read the context by natural language. Details of these templates are discussed in the remaining section.

**Opinion-based prompts.** We propose to transform the context to a narrator's statement and the question to enquire about the narrator's opinion in this statement. This approach is motivated by our own cognitive process for answering different types of questions. When answering questions that seek factual information, we can often rely on our own memory and answer without needing to refer to the context, as these questions typically have only one correct answer. However, when questions are seeking opinions from someone else (in this context, the narrator), it is important to comprehend the narrator's words before answering the questions, as opinions may vary from person to person. Besides, as opinions are inherently subjec-

---

[3]Options only apply to multiple-choice tasks and are removed in free-form text generation tasks.

tive and can be influenced by many factors such as personal experiences and beliefs, opinion-seeking questions are sometimes difficult to answer solely based on the narrator's statement compared to a fact-seeking question that typically has definite and verifiable answer(s). As a result, transforming factual questions into opinion-seeking questions can lead to more attention to the context, as memorized answers alone may not suffice. It also helps the model more selectively predict under cases where contexts do not describe answers. Both factors lead to improved faithfulness with LLMs. The *opinion-based* prompting template is as follows:

> **Opinion-based prompt**
>
> Bob said, "$\{c\}$" Q: $\{q\}$ in Bob's opinion? Options: $\{o\}$ A:

Throughout our experiments, we consistently use Bob to represent the narrator for the context, although other names could be utilized as well.

**Instructed prompts.** We also explicitly instruct LLMs to read context by natural language. We start by extending questions in prompts with attributive phrases such as "based on the given text", leading to the following *attributed* prompting template:

> **Attributed prompt**
>
> $\{c\}$ Q: $\{q\}$ based on the given text? Options:$\{o\}$ A:

We also augment the prompts with natural language instructions. Since manually writing instructions can be laborious and often fails to account for the compatibility between instructions and LLMs, we leverage automatic prompt engineering (APE; Zhou et al. 2022) to generate the prompts. Using a few instances and their desired outputs as demonstrations, APE uses LLMs to automatically generate candidate instructions and select the best one based on the results on a dev set (see Appx. §A for generated instructions). We then use the following *instruction*-based prompting template:

> **Instruction-based prompt**
>
> Instruction: $\{Instruction\}$  $\{c\}$ Q: $\{q\}$? Options: $\{o\}$ A:

Experiments show that all prompting templates perform better than the base prompting template. Specifically, opinion-based prompts outperform instructed prompts in both knowledge conflict and

prediction with abstention facets, and combining these two prompting methods results in the most significant improvements.

## 3.2  Counterfactual Demonstration

Using demonstrations is a standard way to perform few-shot inference on LLMs (Brown et al., 2020). To enhance the faithfulness of language models in knowledge conflict scenarios, previous studies (Li et al., 2022; Neeman et al., 2022) propose to fine-tune the models using counterfactual instances, where the facts in the context are substituted with false ones, and the model learns to update its predictions accordingly. Following this strategy, we propose to use counterfactual instances as demonstrations for LLMs. To do so, we start with a labeled set of counterfactual instances and a test instance and then use KATE (Liu et al., 2022) to retrieve the most relevant counterfactual instances as demonstrations. We encode both the test instance and counterfactual instances with RoBERTa$_{\text{nli+sts-b}}$ (Liu et al., 2019; Reimers and Gurevych, 2019) and select the top counterfactual instances based on cosine similarity. As a part of our analysis, we also experimented with using the original (factual) instances as demonstrations but found this approach to underperform counterfactual demonstrations and sometimes even zero-shot inference.

## 4  Experiments

This section presents our experimental setups (§4.1) for the evaluation of the proposed methods concerning two aspects of faithfulness: knowledge conflict (§4.2) and prediction with abstention (§4.3). We provide additional analysis (§4.4) on results across different model sizes and results on the original datasets. We also show examples of prompts and LLMs' outputs in the case study (§4.5).

## 4.1  Experimental Setup

Our experiments are conducted using the InstructGPT model (text-davinci-003, 175B parameters) and LLama-2-7B-chat (Touvron et al., 2023). We use the base prompt as our baseline, and compare it against the proposed prompting templates described in §3.1, including attributed prompt (ATTR), instruction-based prompt (INSTR), opinion-based prompt (OPIN), and the combination of opinion-based prompt and instruction-based prompt (OPIN + INSTR). We evaluate the effectiveness of these templates in both zero-shot and

| GPT-3.5 | | MRC | | | | RE | | | |
|---|---|---|---|---|---|---|---|---|---|
| | | $p_s\uparrow$ | $p_o\downarrow$ | $M_R\downarrow$ | EM↑ | $p_s\uparrow$ | $p_o\downarrow$ | $M_R\downarrow$ | $F_1\uparrow$ |
| Zero-shot | Base | 59.0 | 32.1 | 35.2 | 6.2 | 73.9 | 21.5 | 22.5 | 81.0 |
| | Attr | 71.9 | 14.4 | 16.6 | 29.6 | 72.4 | 23.6 | 24.6 | 80.0 |
| | Instr | 74.2 | 16.0 | 17.7 | 27.1 | 75.8 | 15.6 | 17.1 | 81.6 |
| | Opin | 79.4 | 9.8 | 11.0 | 24.9 | 76.0 | 19.6 | 20.5 | 82.9 |
| | Opin + Instr | 79.1 | 7.9 | 9.1 | 48.6 | 79.4 | 15.0 | 15.9 | 84.7 |
| Original | Base | 43.3 | 49.4 | 53.3 | 35.1 | 76.2 | 19.8 | 20.6 | 83.3 |
| | Attr | 54.1 | 37.7 | 41.0 | 45.5 | 76.5 | 19.7 | 20.5 | 83.7 |
| | Instr | 54.6 | 37.7 | 40.8 | 45.8 | 77.3 | 18.4 | 19.2 | 84.2 |
| | Opin | 60.6 | 28.7 | 32.1 | 51.1 | 76.8 | 18.4 | 19.3 | 83.8 |
| | Opin + Instr | 64.7 | 26.8 | 29.3 | 53.8 | 78.2 | 17.1 | 17.9 | 84.9 |
| Counter | Base | 86.9 | 6.5 | 7.0 | 80.2 | 78.7 | 13.7 | 14.8 | 83.9 |
| | Attr | 89.1 | 4.6 | 4.9 | 83.0 | 79.7 | 13.0 | 14.0 | 84.3 |
| | Instr | 86.2 | 6.3 | 6.8 | 80.1 | 78.0 | 12.8 | 14.1 | 82.9 |
| | Opin | 90.1 | 3.7 | 3.9 | 84.3 | 79.7 | 12.8 | 13.8 | 84.4 |
| | Opin + Instr | **90.9** | **2.8** | **3.0** | **85.2** | **80.0** | **10.5** | **11.6** | **85.1** |

Table 1: Results (in %) on GPT-3.5-175B in the knowledge conflict setting. The overall best results are highlighted in **bold**. The best and the second best results in each setting are highlighted in green and orange, respectively.

| LLama-2 | | MRC | | | | RE | | | |
|---|---|---|---|---|---|---|---|---|---|
| | | $p_s\uparrow$ | $p_o\downarrow$ | $M_R\downarrow$ | EM↑ | $p_s\uparrow$ | $p_o\downarrow$ | $M_R\downarrow$ | $F_1\uparrow$ |
| Zero-shot | Base | 50.8 | 40.9 | 44.6 | 3.5 | 15.3 | 67.6 | 81.6 | 12.8 |
| | Attr | 66.2 | 23.8 | 26.4 | 4.7 | 13.2 | 66.5 | 83.5 | 10.9 |
| | Instr | 77.7 | 19.7 | 20.2 | 27.0 | 19.6 | 9.2 | 75.1 | 13.2 |
| | Opin | 74.6 | 14.6 | 16.4 | 9.4 | 20.7 | 63.4 | 75.4 | 14.4 |
| | Opin + Instr | 77.8 | 13.9 | 15.1 | 13.7 | 21.6 | 57.9 | 72.8 | 11.8 |
| Original | Base | 56.7 | 39.7 | 41.1 | 19.4 | 27.6 | 62.3 | 69.4 | 9.4 |
| | Attr | 61.7 | 34.5 | 35.9 | 25.2 | 29.4 | 58.9 | 66.7 | 11.2 |
| | Instr | 59.4 | 35.7 | 37.5 | 25.5 | 34.6 | 53.6 | 60.8 | 13.2 |
| | Opin | 67.1 | 32.1 | 32.4 | 18.5 | 32.2 | 57.1 | 63.9 | 10.9 |
| | Opin + Instr | 70.6 | 26.8 | 27.5 | 27.6 | 35.7 | 51.3 | 59.0 | 11.5 |
| Counter | Base | 84.4 | 7.8 | 8.4 | 39.2 | 76.3 | 14.8 | 16.2 | 38.9 |
| | Attr | 85.9 | 7.0 | 7.6 | 44.1 | 76.5 | 14.2 | 15.7 | 39.5 |
| | Instr | 85.5 | 6.7 | 7.3 | 47.1 | 76.0 | 14.4 | 15.9 | 37.3 |
| | Opin | 86.7 | 6.2 | 6.7 | 38.1 | 76.3 | **13.8** | **15.4** | **41.7** |
| | Opin + Instr | **88.1** | **4.9** | **5.2** | **49.6** | **77.3** | 14.2 | 15.5 | 36.9 |

Table 2: Results (in %) on LLama-2-7B-chat in the knowledge conflict setting. The overall best results are highlighted in **bold**. The best and the second best results in each setting are highlighted in green and orange, respectively.

few-shot settings (with demonstrations).

## 4.2 Knowledge Conflict

**Datasets.** We evaluate in the knowledge conflict setting using counterfactual datasets that contain incorrect facts, which can conflict with what the LLM has memorized. We use two datasets based on real-world texts: natural questions (Kwiatkowski et al., 2019) for MRC and Re-TACRED (Stoica et al., 2021) for relation extraction (RE). To create counterfactuals, we adopt the framework proposed by Longpre et al. (2021), which modifies the context to support a counterfactual answer. Specifically, for MRC, we follow Longpre et al. (2021)

and replace the gold entity answer in the context with a randomly sampled entity of the same entity type from the corpus. For RE, we first randomly sample a context that has the entity mentions of the same type but different relations from the original one, and then insert the original entities into the sampled context. In this scenario, a faithful LLM should update its prediction to the new answer instead of returning the original one. Moreover, to measure LLMs' ability to update answers, we need to ensure that they have memorized the knowledge of the original answers in the first place. Therefore, we only evaluate LLMs on a subset of instances on which these models can correctly predict the

original answers without additional contexts.

**Task setup.** We use the same set of evaluation metrics as Longpre et al. (2021). Specifically, we measure the frequency that the LLMs' predictions *contain* an exact match of the original answers ($p_o$) and the substituted answers ($p_s$), after both predictions and answers have been normalized by removing stop words and punctuation To assess the model's reluctance to update its prediction, we use the memorization ratio ($M_R$), which is calculated as $M_R = \frac{p_o}{p_o + p_s}$. A completely faithful LLM should have an $M_R$ of 0. We also report task-specific metrics, including exact match (EM) for MRC and $F_1$ for RE. For EM, we also use normalized predictions and answers, but the requirement is that the prediction and answer must be exactly the same, rather than just containing the answer. We conduct experiments in three different settings: zero-shot, demonstration using original instances, and demonstration using counterfactual instances. We retrieve demonstrations from the original/counterfactual training set, and evaluate LLMs on the counterfactual test set. In the few-shot setting, we utilize a maximum of 16 demonstration instances, up to the limit of the LLM's context window.

**Results and discussion.** The results in Tab. 1 and Tab. 2 demonstrate that the combination of OPIN + INSTR prompting and counterfactual demonstrations is generally the most effective. Compared to the zero-shot base prompts, there is a reduction of 32.2% in $M_R$ for MRC and a 10.9% reduction for RE on GPT-3.5. Similarly, on LLaMA-2-7B-chat, there is a 39.4% reduction in $M_R$ for MRC and a 57.3% reduction for RE. We also find that opinion-based prompts generally perform better than other templates, achieving the second-best results on 17 out of 24 metrics on GPT-3.5, and 9 out of 24 metrics on LLama-2, indicating that LLMs are more faithful to the context when answering opinion-seeking questions. Combining opinion-based prompts and instruction-based prompts further improves faithfulness, with the best results obtained in 23 out of 24 metrics on GPT-3.5, and 19 out of 24 metrics on LLama-2.

When it comes to few-shot settings, counterfactual demonstrations lead to further improved performance. Using the original (factual) instances as demonstrations, on the other hand, leads to limited effects or may even impair faithfulness in MRC. This finding suggests that demonstrations do not always improve the generalization of LLMs' inference, especially when they contain dataset bias. In the MRC experiments, the natural questions dataset used is constructed based on Wikipedia, which mainly consists of world facts. This potentially allows for a simplicity bias of LLMs where questions can be answered without contexts. Therefore, our study suggests the importance of using counterfactual demonstrations in knowledge conflict scenarios.

### 4.3 Prediction with Abstention

**Datasets.** As for the second aspect of faithfulness, we evaluate LLMs' ability to selectively abstain from making uncertain predictions based on irrelevant context. Since existing datasets such as SQuAD 2.0 (Rajpurkar et al., 2018) generally contain questions with confusion (Yatskar, 2019) and are less related to our problem setting, we curate our own evaluation data based on RealTime QA (Kasai et al., 2022), a dataset that inquires about novel information from June 2022 onwards. In this formulation, LLMs are presented with a question and multiple choices, and they need to choose the correct answer based on several retrieved documents. These documents were obtained using tools like Google custom search and may not contain the answer to the question. To adapt this dataset to our setting, we added a new "I don't know" choice and relabeled the dataset. Instances where the retrieved documents do not answer the question are relabeled to "I don't know". We used questions in the first six weeks of 2022 as the test set and randomly picked three questions of 2023 as demonstration instances. This process results in a total of 113 test instances, including 63 answerable questions and 50 unanswerable ones.

**Task setup.** We calculate the probability of a choice as $P(\text{choice}|\text{prompt})$ followed by normalization across all choices.[4] We report accuracy on the entire dataset (All), accuracy on the subset of questions that can be answered based on retrieved documents (HasAns), and accuracy on questions that cannot be answered based on retrieved documents (NoAns). The latter two metrics measure LLMs' ability to extract answers from context and

---

[4]We tried three methods to calculate $P(\text{choice}|\text{prompt})$: joint probability, per-token probability (joint probability normalized by length), and unconditional probability as done in Brown et al. (2020). We find that joint probability works the best for GPT-3.5, while per-token probability works the best for LLama-2.

| | Method | GPT-3.5 | | | LLama-2 | | | |
| | | Acc↑ | | Brier↓ | | Acc↑ | | Brier↓ |
| | | NoAns | All | | HasAns | NoAns | All | |
|---|---|---|---|---|---|---|---|---|
| Zero-shot | Base | 30.6 | 68.5 | 29.4 | 88.7 | 14.3 | 55.9 | 30.0 |
| | Attr | 65.3 | 84.4 | 14.6 | 87.1 | 16.3 | 55.9 | 29.7 |
| | Instr | 81.6 | 91.7 | 7.7 | **91.9** | 26.5 | 63.1 | 27.4 |
| | Opin | 83.3 | 92.6 | 6.6 | 85.5 | 30.6 | 61.3 | 27.8 |
| | Opin + Instr | 87.8 | 94.4 | 5.2 | 88.7 | 36.7 | 65.8 | 27.4 |
| Few-shot | Base | 73.5 | 88.2 | 11.2 | 56.5 | 69.4 | 62.2 | 27.9 |
| | Attr | 81.6 | 91.9 | 8.0 | 59.7 | 67.3 | 63.1 | 27.0 |
| | Instr | 85.7 | 93.7 | 6.1 | 51.6 | 81.6 | 64.9 | 26.6 |
| | Opin | 87.8 | 94.6 | 4.1 | 50.0 | 87.8 | **66.6** | 26.6 |
| | Opin + Instr | **89.8** | **95.5** | **3.4** | 43.5 | **91.8** | 64.9 | **26.0** |

Table 3: Results (in %) for GPT-3.5 and LLama-2 on RealTime QA. The overall best results are highlighted in **bold**. The best and the second best results in each setting are highlighted in green and orange, respectively. As all prompts achieve perfect accuracy (100%) on the **HasAns** subset for GPT-3.5, it is not included in the table.

their ability to abstain from making predictions when the context does not describe the answer, respectively. Besides, we use the probability of "I don't know" as LLM's probability estimation of whether the question can be answered. We use the Brier score to evaluate the accuracy of the estimation, which measures the mean squared difference between the estimation and the true binary outcome of answerability. We use three demonstrations for each instance in the few-shot setting, where some instances are filtered out during evaluation due to exceeding the context length of LLMs.

**Results and discussion.** The results detailed in Tab. 3 reveal that the OPIN + INSTR prompt outperforms all others on GPT-3.5, both in the zero-shot and few-shot settings, surpassing base prompts by 57.2% and 16.3% in NoAns subset accuracy, respectively. For LLama-2, this approach similarly outperforms base prompts by 22.4% in both settings. Furthermore, the Brier score is reduced by 24.2% and 7.8% compared to base prompts for GPT-3.5 in the two settings, respectively, and by 2.6% and 1.9% on LLama-2. The OPIN prompt is the second best in terms of these metrics. These findings demonstrate that opinion-based prompts can enhance the LLMs' ability to make selective predictions. In addition, The use of demonstrations consistently improves the LLMs' ability to make selective predictions, as evidenced by the lower Brier scores and higher NoAns accuracy in the few-shot setting compared to the zero-shot setting.

## 4.4 Additional Analysis

**Memorization by different sizes of LLMs.** Fig. 2 shows the memorization ratio $M_R$ across different

sizes of InstructGPTs under the zero-shot evaluation of natural questions.[5] Overall, OPIN + INSTR consistently outperforms other prompts across different model sizes. In the upper plot, results are shown for filtered evaluation sets where the corresponding LLMs can correctly predict the original answers without additional contexts, thereof the size of evaluation sets varies across different LLMs.[6] We observe that $M_R$ generally decreases with increased model size, showing that larger LLMs are better at updating memorized answers based on given contexts in knowledge conflicts. However, the lower plot reveals that larger LLMs have more severe memorization on the full (unfiltered) evaluation set. This is because larger LLMs can memorize more answers than smaller ones, as evidenced by the number of instances in the filtered evaluation set where larger LLMs have more instances. Our analysis suggests that while larger LLMs are better at updating memorized answers, they still tend to have more memorization due to the larger number of memorized answers. Therefore, we need to pay more attention when using larger LLMs in scenarios with new or potentially conflicting knowledge.

**Selective prediction by different sizes of LLMs.** Fig. 3 shows the Brier score across different sizes of InstructGPTs under the zero-shot evaluation of RealTime QA. On smaller LLMs, opinion-based prompt achieves similar or even higher Brier score

---

[5]The 0.3B, 1.3B, 6.7B models refer to text-ada-001, text-babbage-001, text-curie-001, respectively. We do not perform few-shot evaluation as different sizes of LLMs have different maximum input lengths and can take different numbers of demonstrations, thus hard to be compared to each other.

[6]The sizes of the filtered evaluation sets, in the order of increased model sizes, are 121, 132, 756, and 2,773.

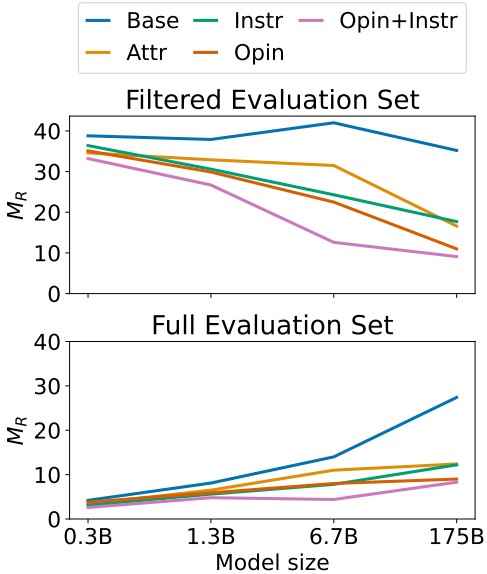

Figure 2: Memorization ratios across different sizes of InstructGPTs, evaluated in the zero-shot setting.

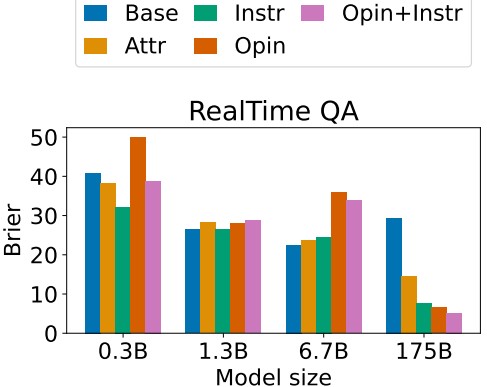

Figure 3: Brier scores across different sizes of Instruct-GPTs in the zero-shot setting of RealTime QA.

| | Method | $p_o\uparrow$ | EM$\uparrow$ |
|---|---|---|---|
| Zero-shot | Base | 92.1 | 11.1 |
| | Opin | 91.3 | 25.2 |
| | Opin + Instr | 90.5 | 57.2 |
| Original | Base | 93.2 | 77.8 |
| | Opin | 92.7 | 78.7 |
| | Opin + Instr | 93.9 | 80.1 |
| Counter | Base | 93.6 | 82.1 |
| | Opin | 92.8 | 82.3 |
| | Opin + Instr | 92.7 | 82.1 |

Table 4: Results (in %) on the filtered evaluation set of natural questions with original contexts and answers.

than base prompts, indicating it does not improve the selective prediction ability of LLMs. We hypothesize that this is because smaller LLMs have inferior reading comprehension ability, resulting in uncertainty in many instances. Opinion-based prompts change uncertain predictions of answerable questions to *I don't know*, which could lead to worse results. For other prompting templates, we do not observe a consistent improvement across different LLMs either. This analysis shows that while the selective prediction ability can be more easily activated by zero-shot prompting for LLMs such as text-davinci-003, smaller LLMs may require dedicated adaptations such as calibration and finetuning to activate this ability.

**Results on original datasets.** While our main experiments demonstrate the effectiveness of the proposed methods in resolving knowledge conflicts, LLMs in real-world applications may also see instances without knowledge conflicts. Therefore, we investigate how our methods affect inference when the memorized answers align with the given contexts. To do so, we evaluate LLMs on the same set of filtered evaluation set used in the main results section (§4.2), but we use the original contexts and answers instead of counterfactual ones. The results in Tab. 4 show that opinion-based prompts yield similar or better results in all settings. Furthermore, using either counterfactual or original demonstrations does not significantly impact results on the original (factual) dataset. This analysis

reveals that our methods do not impair performance on instances without knowledge conflicts.

### 4.5 Case Study

Tab. 5 shows some examples of prompts and the corresponding answers generated by text-davinci-003. The left column of the table presents a knowledge conflict case where the original answer, *Lady Gaga*, is replaced with a counterfactual answer, *Bosco*. When using base prompts, LLM ignores the context and return the memorized answer *Lady Gaga*. However, using opinion-based prompts and their combination with instructions leads to a more faithful response, with the language model returning *Bosco* in the given context. The right column presents a scenario where the retrieved context from Google search is irrelevant to the given question. In such cases, base prompts still return a choice, leading to a potentially incorrect answer. However, opinion-based prompts and their combination with instructions can abstain from making predictions and return *I don't know*. These examples demonstrate the effectiveness of proposed

| | Knowledge Conflict | Prediction with Abstention |
|---|---|---|
| **Context** | (Counterfactual passage) The Super Bowl LI Half-time show took place on February 5, 2017, at NRG Stadium in Houston, Texas as part of Super Bowl LI. The show was headlined by **Bosco**, who performed a medley of her songs, including newer material from her most recent studio album Joanne. | Tara Connolly is senior gas campaigner at Global Witness, an international NGO working towards a more sustainable, just and equal planet. She has over a decade of experience in EU energy policy. The views expressed in this commentary are her own. |
| **Prompt** | Instruction: read the given information and answer the corresponding question. Bob said, "The Super Bowl ... album Joanne." Q: who performed the halftime show at Super Bowl 51 in Bob's opinion based on the given text? | Instruction: answer a question based on the provided input-output pairs. Bob said, "Tara Connolly ... are her own." Q: Mo Farah made public that he was trafficked from which African country to the UK in Bob's opinion based on the given text? Choices: Somaliland; Djibouti; Ethiopia; Somalia; I don't know |
| **Base** | Lady Gaga ✗ | Somalia ✗ |
| **Attr** | Lady Gaga ✗ | Somalia ✗ |
| **Instr** | Lady Gaga ✗ | Somaliland ✗ |
| **Opin** | Bosco ✓ | I don't know ✓ |
| **Instr + Opin** | Bosco ✓ | I don't know ✓ |
| **Answer** | Bosco | I don't know |

Table 5: Examples of prompts and LLMs' corresponding predictions. In the "Prompt" row, we show and highlight the added parts from different prompting templates including attributed prompts, instruction-based prompts, and opinion-based prompts.

prompts in generating context-faithful responses.

## 5 Conclusion

In this paper, we focus on addressing the faithfulness issue of LLMs in context-specific NLP tasks, particularly in scenarios with knowledge conflict and prediction with abstention. We propose that two methods, opinion-based prompts and counterfactual demonstrations, are effective in improving LLMs' faithfulness to contexts. We evaluate our methods on three datasets of two tasks, namely machine reading comprehension and relation extraction, and observed significant improvement in faithfulness to contexts. Future work includes evaluating the effectiveness of proposed methods on a broader range of NLP tasks such as open-domain QA and summarization, and studying other techniques to improve faithfulness further.

## Acknowledgement

We appreciate the reviewers for their insightful comments and suggestions. Wenxuan Zhou and Muhao Chen are supported by the NSF Grant IIS 2105329 and the DARPA MCS program under Contract No. N660011924033 with the United States Office Of Naval Research.

## Limitations

In this study, our main focus is on the utilization of context-augmented prompting, assuming the reliability of the provided context. However, real-world scenarios can be more complicated, which may involve retrieved contexts that contain erroneous or conflicting information. Assessing the factuality of the context solely based on the provided information becomes challenging, as it depends on additional factors such as trustworthiness and timeliness of the information source. Due to the complexity and challenges associated with verifying context reliability, we do not address this issue within the scope of this work. Furthermore, it is important to note that our paper primarily concentrates on the capability of LLMs to generate updated answers or decisions for given questions, rather than exploring more intricate tasks that require the model to apply the updated knowledge in multi-hop reasoning.

## Ethical Considerations

Due to the availability of test data, the experiments conducted in this work has been in English, while future work can consider extending the use of proposed techniques to tasks in other languages. The

datasets used in this work are public datasets that may not be free of inherent biases. However, the introduced context-faithful prompting techniques in this work do not introduce additional biases beyond what the data have presented.

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

# Appendices

## A    Settings of Automatic Prompt Engineering

We run APE using their official code[7] and default hyperparameters. In the knowledge conflict setting, we use counterfactual datasets to generate instructions. While the APE paper recommends using instructions generated by the same model in inference, we find that smaller LLMs do not generate meaningful instructions for our datasets. Therefore, we use instructions generated by text-davinci-003 across different scales of LLMs in additional analysis. The top three instructions generated by APE on each dataset are listed below. We use the top one instruction in experiments.

**Natural questions:**

1. read the given information and answer the corresponding question.

2. read a piece of text and then use the information in the text to answer a question.

3. "Read the given information and answer the questions that follow."

**Re-TACRED:**

1. identify the relationship between two entities from a list of options.

2. identify the relationship between two entities based on the given input-output pairs.

3. identify the relationship between two entities given the input-output pairs.

**RealTime QA:**

1. answer a question based on the provided input-output pairs.

2. ask a question with a set of choices and ask the friend to provide the correct answer.

3. answer a question related to a news article.

---

[7]https://github.com/keirp/automatic_prompt_engineer