# OpenReview forum: "Context-faithful Prompting for Large Language Models"
_EMNLP/2023/Conference — EMNLP 2023 Findings_

### Official Review · Reviewer_RhhF · 2023-08-04

**Typos Grammar Style And Presentation Improvements:** None that I could find
**Soundness:** 4

**Excitement:**

3: Ambivalent: It has merits (e.g., it reports state-of-the-art results, the idea is nice), but there are key weaknesses (e.g., it describes incremental work), and it can significantly benefit from another round of revision. However, I won't object to accepting it if my co-reviewers champion it.

**Paper Topic And Main Contributions:**

The authors provide a detailed analysis of utilizing instruction based and opinion-based formulation prompts for the purpose of contextual question answering. The prompts that the authors propose are simple and can be utlizied in various language models. More importantly, the authors demonstrate that this style of prompting tends to force the model to leverage the context given rather than recall the answer from training data.

**Questions For The Authors:**

Why only utilize the InstrucGPT family of models and not something else?
Why not expand on the use of auto prompting to discover a stronger baseline?

**Reasons To Accept:**

The proposed methods are simple (the prompts can be easily formulated), and the results indicate that the formulations can force the models to rely more on the context that they are given rather than the prior knowledge that they may have. These methods are crucial as they enable the models to rely more on context, rather than requiring retraining or extensive fine-tuning, making them more cost-effective and independent of specific models. The paper is also well written and easy to follow.

**Reasons To Reject:**

While the prompting strategy seems effective, the work seems to be missing more baselines
* If the authors are testing this in large language models, I would use other publicly available models such as LLaMa or T5 etc. to verify that the prompting strategies work across different types of training data and are not exclusive to the result of the InstructGPT models
* I would also add a baseline on a fully auto engineered prompt. The authors mention in line 274, that they use auto prompt engineering. I would put another baseline in which the full prompt save for {c}, {q} is auto engineered and just check the results of this. Because while the authors formulate that the instruction/opinion prompts provide improved context usage and their combination does too, it may be that some other style of prompting that is discovered with the auto prompt engineering may be even better/serve as a strong baseline.

**Reproducibility:**

5: Could easily reproduce the results.

**Reviewer Confidence:**

4: Quite sure. I tried to check the important points carefully. It's unlikely, though conceivable, that I missed something that should affect my ratings.

---

> ### Author Rebuttal · Authors · 2023-08-29
>
> Thank you for the thorough review and valuable insights into our work.
>
> **Use of Other Language Models**:
>
> We acknowledge the importance of considering multiple baselines and using different language models to validate our approach's effectiveness. We will add results on other LMs such as T5, Alpaca and ChatGPT.
>
> **Full Auto Engineered Prompt Baseline**:
>
> Your recommendation to include a fully auto-engineered prompt as a baseline is valid. However, it's important to note that the current auto-prompting techniques we use only produce instructions based on given input-output pairs. At present, we are not aware of auto-prompting techniques that can also modify context and queries. Still, we value this feedback and will ensure to provide a more detailed discussion regarding this in our revised paper.

---

### Official Review · Reviewer_FEGa · 2023-08-04

**Typos Grammar Style And Presentation Improvements:** NA
**Soundness:** 3

**Excitement:**

4: Strong: This paper deepens the understanding of some phenomenon or lowers the barriers to an existing research direction.

**Missing References:**

NA

**Paper Topic And Main Contributions:**

This paper focus on resolving the faithfulness issue of the Large Language Model in context specific NLP tasks. The LLM faithfulness is concerned with the validity and reliability of its extraction process, and this work assesses and enhances LLMs’ contextual faithfulness in two aspects: knowledge conflict and prediction with abstention. The research proposes two approaches which are opinion-based prompts and counter-factual demonstrations, are exhibited effectiveness in enhancing LLMs’ faithfulness to contexts.

**Questions For The Authors:**

Question A: More LLMs (i.e., ChatGPT or Alpaca) should be used to validate the effectiveness of the proposed tailored prompt method for convincing the readers.

Question B:  More ablation studies should be performed on the in-context learning demonstration number as some studies have proof that the large demonstration number may not receive a good result across various tasks [1-3].

[1] Li, M., Gong, S., Feng, J., Xu, Y., Zhang, J., Wu, Z., & Kong, L. (2023). In-context learning with many demonstration examples.

[2] Min, S., Lyu, X., Holtzman, A., Artetxe, M., Lewis, M., Hajishirzi, H., & Zettlemoyer, L. (2022). Rethinking the role of demonstrations: What makes in-context learning work?

[3] Chan, C., Cheng, J., Wang, W., Jiang, Y., Fang, T., Liu, X., & Song, Y. (2023). Chatgpt evaluation on sentence level relations: A focus on temporal, causal, and discourse relations.

**Reasons To Accept:**

* This paper investigates a critical and interesting issues of LLM (i.e., faithfulness issue) and focuses on two aspects: knowledge conflict and prediction with abstention. This faithfulness issue from LLM may interest the community.
* This paper proposes two methods and receives a promising performance across three NLP tasks.
* The motivations of the paper relative to those elements are fair and well explained.


**Reasons To Reject:**

*  More LLMs (i.e., ChatGPT or Alpaca) should be used to validate the effectiveness of the proposed tailored prompt method for convincing the readers.
*  More ablation studies should be performed on the in-context learning demonstration number as some studies have proof that the large demonstration number may not receive a good result across various tasks [1-3].

[1] Li, M., Gong, S., Feng, J., Xu, Y., Zhang, J., Wu, Z., & Kong, L. (2023). In-context learning with many demonstration examples.

[2] Min, S., Lyu, X., Holtzman, A., Artetxe, M., Lewis, M., Hajishirzi, H., & Zettlemoyer, L. (2022). Rethinking the role of demonstrations: What makes in-context learning work?

[3] Chan, C., Cheng, J., Wang, W., Jiang, Y., Fang, T., Liu, X., & Song, Y. (2023). Chatgpt evaluation on sentence level relations: A focus on temporal, causal, and discourse relations.

**Reproducibility:**

4: Could mostly reproduce the results, but there may be some variation because of sample variance or minor variations in their interpretation of the protocol or method.

**Reviewer Confidence:**

3: Pretty sure, but there's a chance I missed something. Although I have a good feel for this area in general, I did not carefully check the paper's details, e.g., the math, experimental design, or novelty.

---

> ### Author Rebuttal · Authors · 2023-08-29
>
> We would like to sincerely thank the reviewer for their thoughtful and constructive review of our paper.
>
> Question A: We appreciate your suggestion to validate the effectiveness of our tailored prompt method using additional LLMs such as ChatGPT or Alpaca. We will conduct experiments with these models to provide a more comprehensive evaluation and ensure the broader applicability of our approach in revision.
>
> Question B: Thank you for raising the concern about the number of in-context learning demonstrations. In our paper, we use the maximum number of in-context examples that fit in the context length, resulting in 16 for knowledge conflict and 3 for prediction with abstention. We agree on the importance of ablation studies in fine-tuning this parameter. We will thoroughly investigate the optimal number of demonstrations in revision, considering the insights provided references.

---

### Official Review · Reviewer_8jFq · 2023-08-05

**Soundness:** 3

**Excitement:**

3: Ambivalent: It has merits (e.g., it reports state-of-the-art results, the idea is nice), but there are key weaknesses (e.g., it describes incremental work), and it can significantly benefit from another round of revision. However, I won't object to accepting it if my co-reviewers champion it.

**Paper Topic And Main Contributions:**

LLMs‘ reliance on parametric knowledge can lead to incorrect predictions in context-sensitive NLP tasks. The authors propose two methods, namely opinion-based prompts and counterfactual demonstrations, to assess and enhance LLMs' contextual faithfulness. These methods are can improve the models' ability to handle knowledge conflict without additional training. The experiments conducted on three datasets for two standard NLP tasks show promising results.

**Reasons To Accept:**

The paper addresses a pressing concern in the NLP community regarding the contextual faithfulness of Large Language Models.

**Reasons To Reject:**

While the paper presents experiments on three datasets. It would be valuable to see a broader evaluation on a more diverse set of NLP tasks to evaluate the generalizability of the proposed methods.

**Reproducibility:**

4: Could mostly reproduce the results, but there may be some variation because of sample variance or minor variations in their interpretation of the protocol or method.

**Reviewer Confidence:**

1: Not my area, or paper was hard for me to understand. My evaluation is just an educated guess.

---

> ### Author Rebuttal · Authors · 2023-08-29
>
> Thank you for your insightful comments. We agree that evaluating the proposed methods on a more diverse set of NLP tasks could be beneficial. The choice of the current datasets was guided by the availability and relevance of our focus of study. Specifically, we chose MRC as a widely evaluated task in previous studies on knowledge conflict study, and relation extraction as another representative context-specific NLU task. In light of your feedback, we are willing to extending our evaluations to more tasks, such as text summarization and machine translation, upon releasing of the complete version of our code.

---

### Meta-Review · Area_Chair_5qqB · 2023-09-17

**Recommendation:** 4

**Metareview:**

The reviewers agree this work tackles an important problem, proposes interesting new methods, and achieves promising results. The authors promise to add new experiments that address the primary concerns, primarily around the scope of models and datasets, which would extend the generalizability of the conclusions, and would improve this work's appeal as a contribution worthy of greater discussion in the community.

---

### Decision · Program_Chairs · 2023-10-07

**Decision:**

Accept-Findings

**Comment:**

The reviewers agree this work tackles an important problem, proposes interesting new methods, and achieves promising results. The authors promise to add new experiments that address the primary concerns, primarily around the scope of models and datasets, which would extend the generalizability of the conclusions, and would improve this work's appeal as a contribution worthy of greater discussion in the community.